# Regulation of myoepithelial differentiation

**Renee F. Thiemann**[1¤a], **Scott Varney**[2¤b], **Nicholas Moskwa**[3], **John Lamar**[4], **Melinda Larsen**[3], **Susan E. LaFlamme**[1] *

**1** Department of Regenerative and Cancer Cell Biology, Albany Medical College, Albany, New York, United States of America, **2** Department of Surgery, Albany Medical College, Albany, New York, United States of America, **3** Department of Biological Sciences, University at Albany, State University of New York, Albany, New York, United States of America, **4** Department of Molecular and Cellular Physiology, Albany Medical College, Albany, New York, United States of America

¤a Current address: Catholic Central High School, Troy, New York, United States of America
¤b Current address: Department of Cancer Biology, Thomas Jefferson University, Philadelphia, Pennsylvania, United States of America
* laflams@amc.edu

**Data Availability Statement:** Raw data from our microarray studies can be found on the Gene Expression Omnibus (GEO) repository and listed under GSE197652 and can be accessed at https://

## Abstract

The salivary gland can be permanently impaired by radiation treatment for head and neck cancers. Efforts at tissue regeneration have focused on saliva-producing acinar cells. However, myoepithelial cells are also critical to gland function, but mechanisms that regulate their differentiation are poorly defined. To study myoepithelial differentiation, we employed mSG-PAC1 murine salivary gland epithelial cells. We demonstrate that mSG-PAC1 spheroids exhibit phenotypic plasticity between pro-acinar and myoepithelial cell fates. Increased expression of pro-acinar/acinar or myoepithelial RNAs was identified from spheroids cultured under different media conditions by microarray followed by gene-set enrichment analysis. Spheroids cultured with different medium components expressed proteins typical of either acinar or myoepithelial cells, as detected by immunocytochemistry. We demonstrate that the pattern of TAZ expression in the epithelial compartment of the differentiating murine salivary gland correlates with the expression of the myoepithelial marker alpha-SMA, as is the case for TAZ expression in mSG-PAC1 spheroids. Our analysis also indicates that YAP/ TAZ target genes are upregulated together with myoepithelial markers. Importantly, siRNA targeting of TAZ expression in mSG-PAC1 spheroids diminished the expression of myoepithelial markers. Our results in this in vitro cell model implicate TAZ signaling in myoepithelial differentiation.

## Introduction

Tissue regeneration has become the focus of investigation in multiple contexts and in the salivary gland in particular. Permanent salivary gland damage is a consequence of radiation therapy for head and neck cancers [1–3]. In many cases, damage targets multiple cell types including the saliva-producing acinar cells and myoepithelial cells of the submandibular salivary gland [1–3]. Murine models are currently being employed to understand mechanisms that contribute to regeneration of salivary gland tissue following damage induced by radiation,

www.ncbi.nlm.nih.gov/geo/query/acc.cgi?acc=GSE197652.

**Funding:** This research was funded by the National Institutes of Health (https://www.nih.gov). R01-GM-51540 to S.E.L. from National Institute of General Medical Sciences. R01-DE-027953 to M.L. from the National Institute of Dental & Craniofacial Research. The funders had no role in study design, data collection and analysis, decision to publish, or preparation of the manuscript.

**Competing interests:** The authors have declared that no competing interests exist.

obstruction, or resection, focusing on the submandibular salivary gland [4–9]. Several therapeutic approaches are promising, including transplantation of murine salivary gland stem cells to restore function of damaged glands, as well as tissue engineering methods for the generation of functional organoids [9–13]. Interestingly, lineage tracing experiments in damaged glands demonstrated that cellular plasticity is an important contributor to salivary gland regeneration, and in particular to the expansion of acinar cells to replace those lost due to tissue damage [4, 5, 14, 15].

Important efforts to promote salivary gland regeneration is a knowledge of mechanisms that regulate the differentiation of cell types critical to gland function. Multiple approaches have been used to identify mechanisms that regulate acinar cell differentiation, including mouse genetic models, *ex vivo* organ explants and organoid cultures, as well as three-dimensional (3-D) cell culture [16–22]. Similar approaches have been used to understand the formation of ducts in the developing gland [6, 22–24]. However, in spite of the critical contribution of myoepithelial cells to the secretary function of the salivary gland [25–29], little effort has focused on identifying mechanisms that regulate their differentiation.

To facilitate the identification of pathways that guide myoepithelial differentiation, we developed a three-dimensional (3-D) cell culture system to model myoepithelial differentiation. We employed the pro-acinar murine salivary gland epithelial cells, mSG-PAC1 that we established from P2 murine submandibular salivary glands as we previously described and characterized [18]. We first demonstrated the phenotypic plasticity of mSG-PAC1 between proacinar and myoepithelial cell fates. Using immunofluorescence imaging, qPCR and microarray analyses, together with gene set enrichment analyses, we identified conditions that promote the myoepithelial differentiation of mSG-PAC1 cells and demonstrated a role for the TAZ (*Wwtr1*) transcriptional co-activator in regulating this process.

## Results

### Differentiation towards a myoepithelial phenotype in three-dimensional culture

The murine submandibular salivary gland begins to develop by the invagination of an initial epithelial bud into underlying mesenchymal tissue at embryonic day 12.5 (E12.5) [30]. Multiple rounds of branching morphogenesis follow leading to the arborized structure of the adult gland together with the differentiation of acinar cells that produce saliva and contractile myoepithelial cells that promote the secretion of saliva through ducts to the oral cavity [27, 31, 32]. Recent studies showed that myoepithelial cells differentiate from the outer cuboidal layer of epithelial cells of buds of developing acini [33]. The expression of the myoepithelial marker, α-SMA (*Acta2*) was detectable by E-15.5—E16 [20, 33].

Similar to the outer polarized cuboidal layer of epithelial cells in developing acini of the murine salivary gland, mSG-PAC1 spheroids can be induced to adopt a pro-acinar phenotype, characterized by the establishment of a polarized layer of cuboidal cells at the basal surface of spheroids and by the expression of the pro-acinar marker, aquaporin-5 (AQP5) when cultured in a medium referred to as pro-acinar medium (Pro-A medium) [18]. Because of these similarities, we asked whether we could induce the outer cells of mSG-PAC1 spheroids to differentiate into myoepithelial cells. Others reported that the addition of serum to luminal epithelial cells from the mammary gland can contribute to their differentiation towards a myoepithelial phenotype [34]. To test whether mSG-PAC1 cells could serve as progenitors cells of the myoepithelial cell fate, we compared the phenotype of spheroids cultured for six days in Matrigel with Pro-A medium or pro-myoepithelial (Pro-M) medium (See Materials and Methods for details).

Myoepithelial cells are characterized by their stellate shape with extended processes, flattened nuclear morphology, and expression of smooth muscle proteins important for contraction [35]. While mSGPAC1 spheroids cultured in Pro-A medium maintained an organized structure with outer columnar cells at the basal edge (Fig 1A) as previously reported [18], the cells at the basal surface of spheroids cultured in Pro-M medium adopted morphologies characteristic of myoepithelial cells and expressed the myoepithelial markers, alpha smooth muscle action (α-SMA/*Acta2*) and calponin-1 (*Cnn1*) (Fig 1A). Additionally, these spheroids displayed enhanced expression of α-SMA and the integrin β4 subunit, which is also a myoepithelial marker [36], as well as decreased expression of the pro-acinar marker AQP5 (Fig 1B and 1C). Additionally, cells along the basal surface of spheroids cultured in Pro-M medium have significantly flatter nuclei compared to those cultured in Pro-A medium, quantitated by the increased ratio of nuclear diameter to height (Fig 1E and 1F), which is a characteristic of myoepithelial cells [27, 35]. Altogether, these data suggest that mSG-PAC1 cells exhibit phenotypic plasticity and are capable of differentiating into pro-acinar or myoepithelial phenotype in 3-D culture depending upon culture conditions.

## Transcriptome analysis support the phenotypic plasticity of mSG-PAC1 spheroids

To further explore the physiological relevance of mSG-PAC1 cell plasticity, we employed Clarion S microarrays to identify transcriptional profiles of mSG-PAC1 cells culture as spheroids in either Pro-M or Pro-A medium. The availability of single-cell RNA sequencing data (scRNA-seq) from murine salivary gland myoepithelial cells and acinar cells [37], allowed us to perform GSEA. We found that the myoepithelial gene sets from both developing (P1) and adult glands are significantly enriched in mSG-PA1 spheroids culture in Pro-M medium, whereas the acinar gene sets from developing (P1) and adult glands are enriched in mSG-PAC1 spheroids cultured in Pro-A medium (Fig 2). Additionally, GSEA analysis using scRNA-seq data from the murine mammary gland [38] showed that myoepithelial specific genes are modestly enriched in mSG-PAC1 spheroids cultured in Pro-M medium (S1 Fig in S1 Appendix). Notably, genes specific to luminal progenitor and mature luminal cells are significantly enriched mSG-PAC1 spheroids cultured in Pro-A medium (S1 Fig in S1 Appendix). These analyses further demonstrate the plasticity of mSG-PAC1 cells and support the utility as a discovery tool to identify mechanisms that regulate myoepithelial cell fate decisions.

## Transcriptome analysis reveals enhanced expression of genes involved in processes associated with myoepithelial differentiation, as well as myoepithelial markers, and YAP/TAZ targets

Using the Transcriptome Analysis Console (TAC) from ThermoFisher, we found that the expression of 3870 genes were changed by 2-fold or greater. The expression of 2183 genes were upregulated and 1678 were downregulated in Pro-M compared to Pro-A. The top five hundred genes upregulated in myoepithelial spheroids were compared to hallmark and curated gene sets available from the Molecular Signatures Database (MSigDB) from the Broad Institute. Spheroids cultured in Pro-M medium display enhanced expression of genes associated with processes important to myoepithelial differentiation including tissue morphogenesis (Fig 3A), cell differentiation (Fig 3B), cell projection organization, cytoskeletal organization, and actin filament-based processes, among others (Fig 3C).

Not surprisingly, genes that were significantly upregulated in spheroids cultured Pro-M medium included genes for myoepithelial markers, including α-SMA (*Acta2*) and calponin (*cnn2/cnn3*) and others (Fig 4A). Moreover, these spheroids display a reduction in the

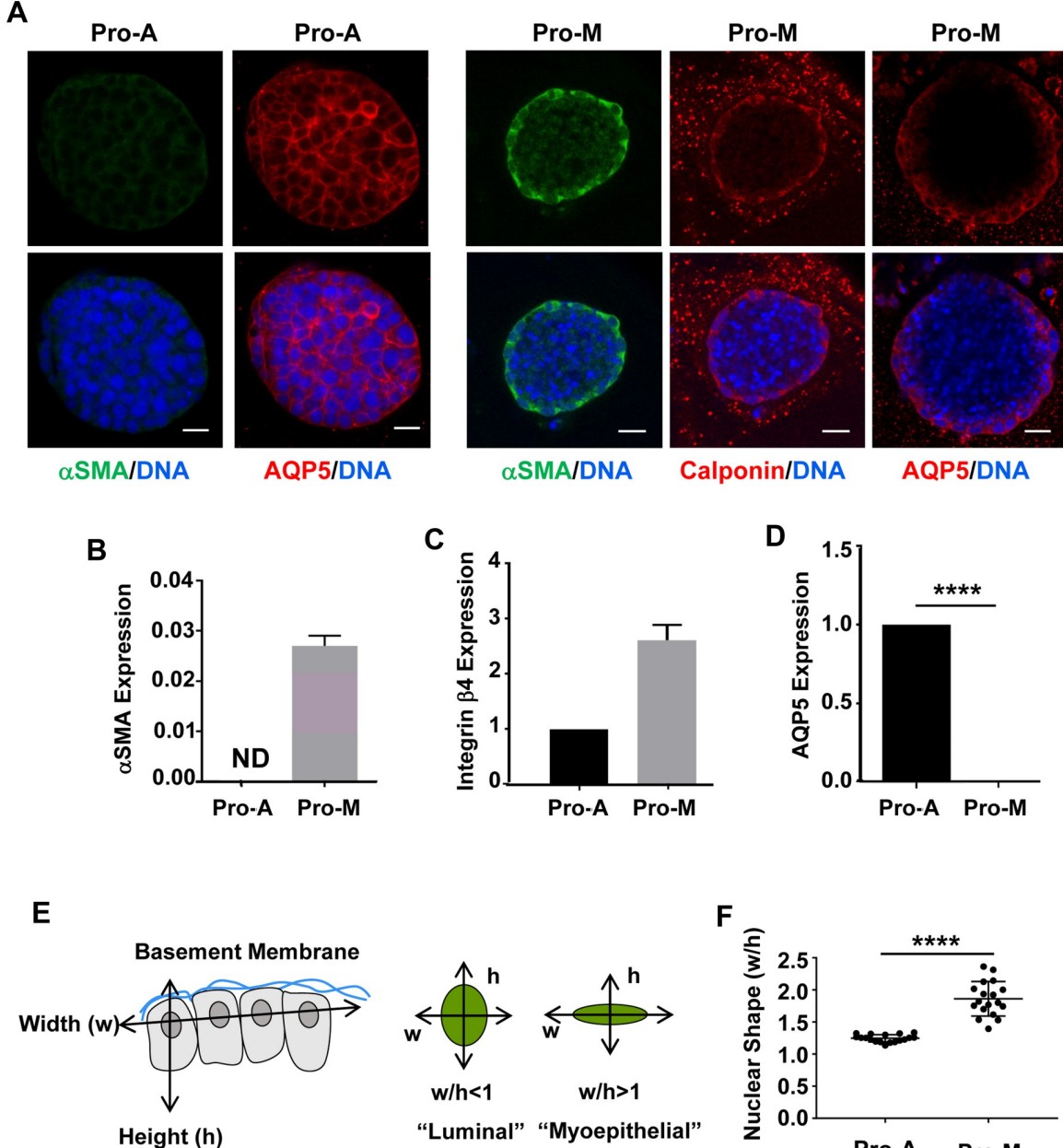

**Fig 1. mSG-PAC1 cells can be induced to express myoepithelial markers and characteristics.** (**A**) Representative confocal images of mSG-PAC1 spheroids cultured for six days in Matrigel in either a medium (Pro-A) that promoted pro-acinar phenotype or a medium (Pro-M) that promoted a myoepithelial phenotype and then immunostained for αSMA, calponin, AQP5 and DNA. Images are maximum projection images of five Z-slices acquired at 40X in 0.4 μm increments. Scale bar, 25 μm. Images are representative of three and two independent experiments for α-SMA and calponin, respectively. (**B-D**) Relative mRNA expression of α-SMA (**B**), the integrin β4 subunit (**C**) or AQP5 (**D**) in mSG-PAC1 cells cultured for six days in Matrigel either in Pro-A or Pro-M medium. Expression is normalized to β-actin and then to expression in Pro-A condition. Data are from three independent experiments and plotted as the mean ± s.e.m. analyzed by Student's T-test. ND = not detected. **$p < 0.01$, ****$p < 0.0001$. (**E, F**) mSG-PAC1 cells cultured in Pro-M medium display significantly flatter nuclei in cells at the basal periphery pf the spheroid. (**E**) Graphical representation of the method used to quantify nuclear shape using ImageJ. (**F**) Spheroids were cultured in either Pro-A or Pro-M medium for 6 days and immunostained for nuclei using DRAQ5. Width and height of nuclei relative to the basement membrane were quantitated in each cell at the basal periphery of spheroids using ImageJ Fiji. Data are from three independent experiments. Plotted is the mean nuclear width/height measured for each of eighteen spheres (n = 18) ± s.e.m. and analyzed by Student's T-test. ****$p < 0.0001$.

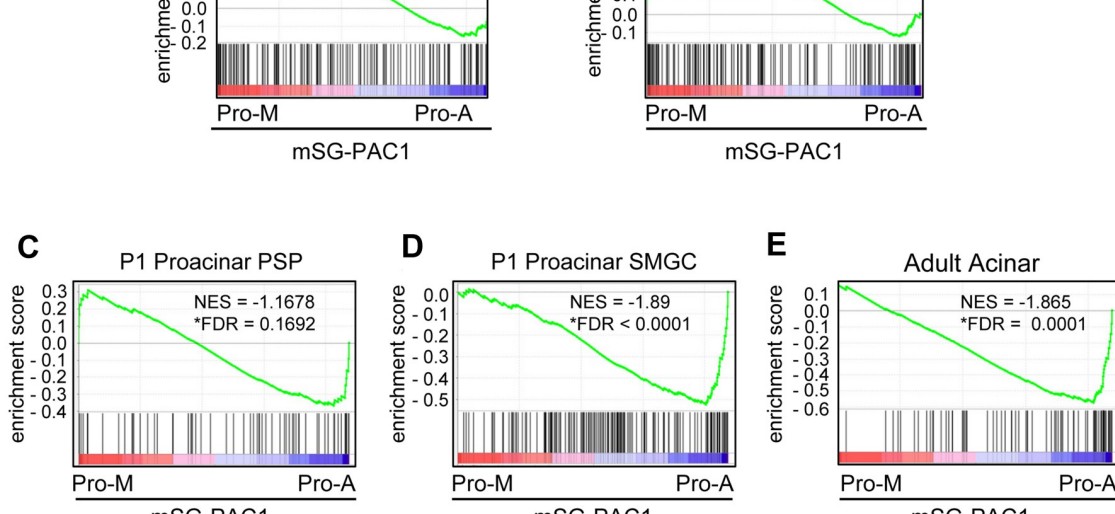

**Fig 2. The culture of mSG-PAC1 spheroids in Pro-M or Pro-A results in the expression of myoepithelial or acinar enriched genes respectively.** GSEA using scRNA-seq gene-sets generated from murine SMG from P1 pups or adult mice [37] showed murine myoepithelial gene upregulation in Pro-M medium and murine acinar gene upregulation in Pro-A medium. **(A, B)** myoepithelial cells from P1 and adult mice, **(C, D)** pro-acinar cells from the *Smgc* and PSP clusters from P1, and **(E)** acinar cells from adult mice. NES = Normalized Enrichment Score and FDR = False Discovery Rate. *FDR q-value < 0.25 represent significant enrichment.

expression of the acinar maturation marker, AQP5 (Fig 4A), which phenocopies the reduction in expression observed by qPCR (Fig 1D). These data suggest that culturing mSG-PAC1 cells in Pro-M medium inhibits their acinar differentiation and promotes differentiation towards a myoepithelial lineage. Interestingly, several genes upregulated in Pro-M medium are established YAP/TAZ target genes, including connective tissue growth factor (*Ctgf*), cysteine-rich protein 61 (*Cyr61*), thrombospondin (*Thbs1*), Ajuba Lim protein (*Ajuba*), and ankyrin repeat domain 1 (*Ankrd1*) (Fig 4B), suggesting YAP/TAZ signaling pathway may be an important regulator of this differentiation event.

## Subcellular localization of TAZ is associated with α-SMA expression during morphogenesis of the SMG

Since many of the genes upregulated in the microarray were canonical target genes of YAP and TAZ, we sought to investigate their role in the differentiation of the myoepithelial layer in mSG-PAC1 cells. YAP/TAZ signaling has been reported to be important in the development and differentiation of branched organs [39–42], and since TAZ has been implicated as a regulator of myoepithelial differentiation in mammary epithelial cells [41], we asked whether TAZ contributed to myoepithelial differentiation in the murine submandibular salivary gland. We first compared the expression and subcellular localization of TAZ in the submandibular gland, at various stages of embryonic development (Fig 5). While TAZ was expressed at E14 and lasted through E19, its expression was increased in acini at E15 and E16 (Fig 5) concurrent with the onset of myoepithelial differentiation [33]. Interestingly, there was also a switch in the subcellular localization of TAZ during this time. While TAZ seemed primarily cytosolic at E14 and E19, its localization appeared nuclear in a subset of suprabasal cells of developing acini at

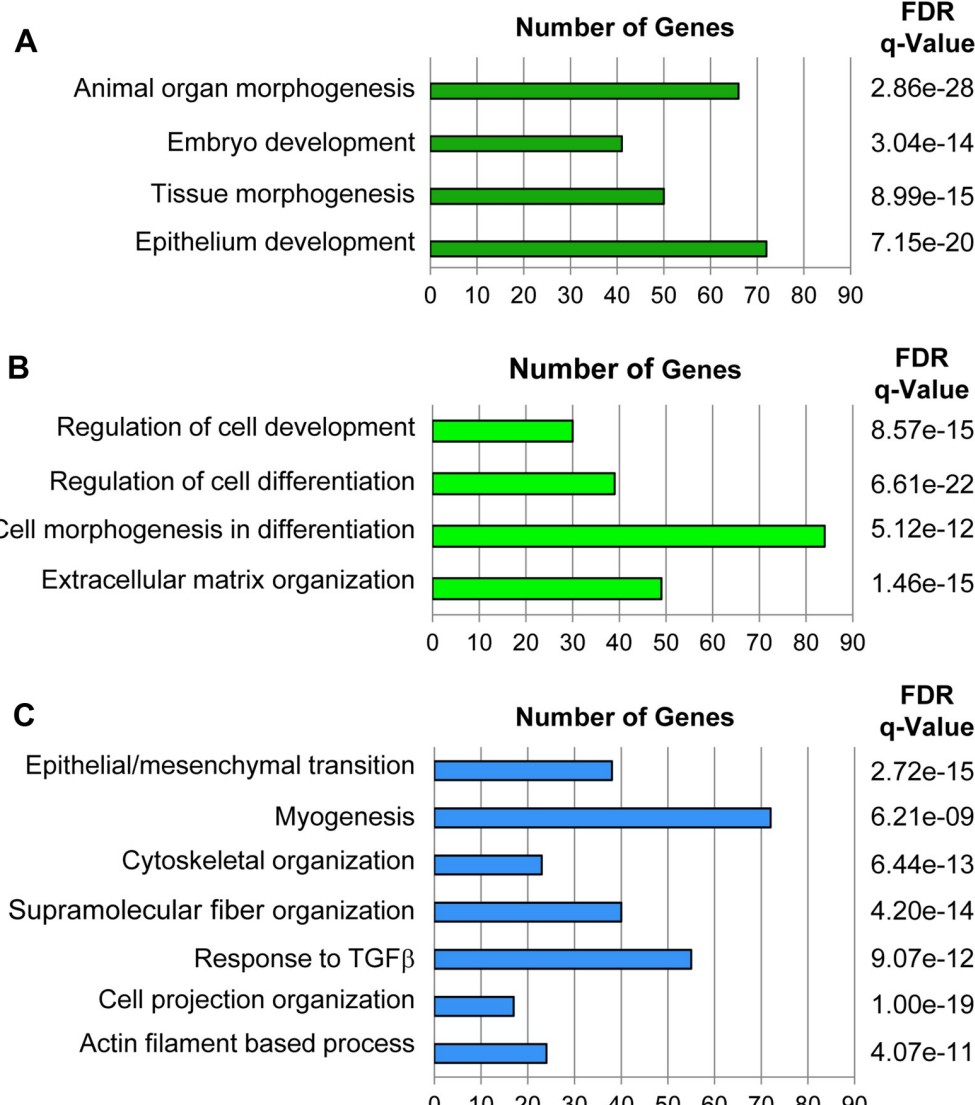

**Fig 3. Genes upregulated in mSG-PAC1 spheroids cultured in Pro-M medium are associated with processes involved in myoepithelial differentiation.** Gene microarrays were performed on RNA isolated from mSG-PAC1 spheroids cultured either in Pro-A or Pro-M medium. N = 3 independent experiments. The top 500 genes most highly expressed by spheroids cultured in the Pro-M medium compared to those expressed in Pro-A M were used to compute overlap with hallmark and curated gene sets from Molecular Signatures Database (MSigDB) from the Broad institute. Genes more highly expressed in the Pro-M conditions are associated with (**A**) tissue, (**B**) cellular, and (**C**) myoepithelial developmental processes.

E15 and E16 (Figs 5 and 6A). Quantitation of TAZ nuclear localization supported this conclusion (Fig 6B). Moreover, this nuclear localization of TAZ was associated with αSMA expression in these cells, suggesting a possible relationship between TAZ activity and myoepithelial differentiation during salivary gland development (Fig 6A).

## TAZ regulates the expression of myoepithelial genes in spheroids

To determine whether TAZ expression is altered when mSG-PAC1 spheroids are cultured for six days in Pro-M medium to promote a myoepithelial-like phenotype, we examined TAZ

**A**

| Gene | Common Name | Fold Change | P-value |
|---|---|---|---|
| *Thbs1* | Thrombospondin | 5674.39 | 1.88e-06 |
| *Acta2* | α-Smooth muscle actin | 1325.38 | 1.92e-09 |
| *Mylip* | Myosin regulatory light chain interacting protein | 8.21 | 0.0003 |
| *Cald1* | Caldesmon | 5.64 | 0.0014 |
| *Cnn2* | Calponin2 | 3.26 | 0.0158 |
| *Cnn3* | Calponin3 | 2.86 | 0.0031 |
| *My110* | Myosin light chain 10 | 1.42 | 0.0334 |
| *Myh10* | Myosin heaving chain 10 | 1.33 | 0.0358 |
| Aqp5 | Aquaporin5 | -5.99 | 0.0015 |

**B**

| Gene | Common Name | Fold Change | P-value |
|---|---|---|---|
| *Thbs1* | Thrombospondin | 5674.39 | 1.88e-06 |
| *Ajuba* | Ajuba Lim protein | 39.23 | 0.0003 |
| *Ctgf* | Connective tissue growth factor | 23.38 | 6.33e-06 |
| *Crim1* | Cysteine-rich transmembrane BMP regulator 1 | 16.75 | 0.0011 |
| *Amotl2* | Angiomotin-like 1 | 10.45 | 0.001 |
| *Tgfb2* | Transforming growth factor, beta2 | 10.13 | 1.3e-06 |
| *Ankrd1* | Ankyrin repeat domain 1 | 9.63 | 0.0005 |
| *Cyr61* | Cysteine-rich protein 61 | 8.37 | 0.0002 |
| *Wwc2* | WW,C2 and c oiled-coil domain containing 2 | 7.74 | 0.0003 |
| *Dlc1* | Deleted in liver cancer 1 | 6.*54* | 0.0003 |

**Fig 4. mSG-PAC1 spheroids cultured in Pro-M medium up regulate myoepithelial markers & YAP/TAZ target genes. (A)** List of myoepithelial genes upregulated in mSG-PAC1 spheroids Pro-M medium cultured in for six days. Data is presented as the fold increase compared to spheroids cultured in Pro-A medium. **(B)** List of YAP/TAZ target genes upregulated in by spheroids cultured Pro-M medium. Data is presented as the fold increase compared to spheroids cultured in Pro-A medium.

expression by immunofluorescence microscopy. The data revealed that TAZ expression was increased in spheroids cultured in Matrigel in Pro-M medium compared with those cultured in Pro-A medium (Fig 7A), with a subset of cells displaying nuclear localization (Fig 7A, insets). Moreover, these spheroids displayed enhanced expression of the canonical YAP/TAZ target genes, CTGF and CYR61 when compared to spheroids cultured in Pro-A medium (Fig 7B and 7C), suggesting that TAZ expression and activity in epithelial cells contributes to the differentiation of the myoepithelial layer.

Our data suggest that the expression and activity of TAZ contributes to the differentiation of the myoepithelial layer in mSG-PAC1 cells cultured in Matrigel. Therefore, we next asked whether the expression of TAZ was required for the expression of myoepithelial markers. We

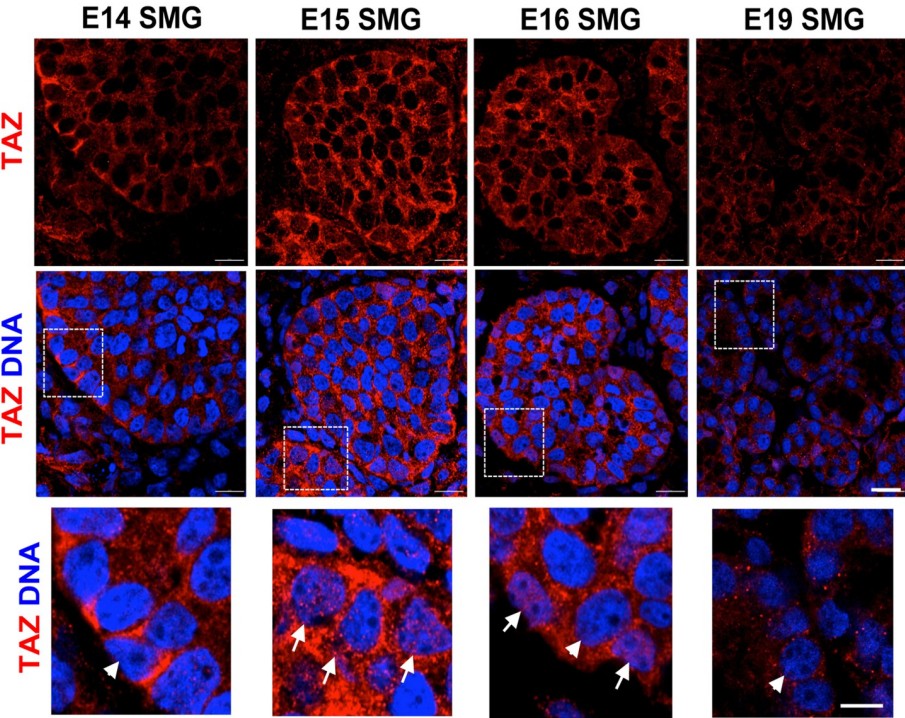

**Fig 5. Expression of TAZ during SMG morphogenesis.** Representative confocal images of embryonic SMGs from E14 through E19 immunostained for TAZ and DNA. Images are a single Z-slice acquired at 63X representative of four SMGs. Scale bar, 10 μm. Insets represent a magnified view of TAZ and DNA around the basal edge of each pro-acinus. Scale bar, 5 μm. White arrows emphasize the co-localization of TAZ and the nuclear marker DRAQ5. White arrowheads emphasize nuclei showing no TAZ co-localization.

treated Matrigel cultures of mSG-PAC1 cells in Pro-M medium with siRNA targeting TAZ. Two distinct siRNA sequences resulted in a significant reduction in TAZ expression levels (Fig 7D and S2 Fig in S1 Appendix). Inhibiting TAZ expression did not significantly alter the expression of YAP or the acinar marker AQP5 (Fig 7D and S2 Fig in S1 Appendix). However, there was a significant reduction in the expression of myoepithelial markers, α-SMA and calponin using two independent siRNAs (Fig 7D and 7E and S2 Fig in S1 Appendix), suggesting that TAZ plays an important role in the transcriptional regulation of these myoepithelial genes. Interestingly, the expression of the integrin β4 subunit was not significantly altered by knockdown of TAZ, suggesting that the expression of the integrin β4 subunit is regulated by a different mechanism.

## Discussion

In our current study, we first demonstrated the phenotypic plasticity of mSG-PAC1 salivary gland epithelial cells. We then used these cells to develop a 3-D cell culture model to study myoepithelial differentiation. Employing this model, we identified a role for the transcriptional co-activator TAZ in contributing to this differentiation process.

The regulation of YAP and/or TAZ activity is important for the proper development and differentiation of many branched organs, including the salivary gland [39–43]. In the developing gland, YAP promotes the transcription of genes that contribute to the expansion of ductal progenitors and the suppression of YAP activity by the LATS kinase, a component of the Hippo pathway, is required for proper ductal maturation [40]. The activity of TAZ in the

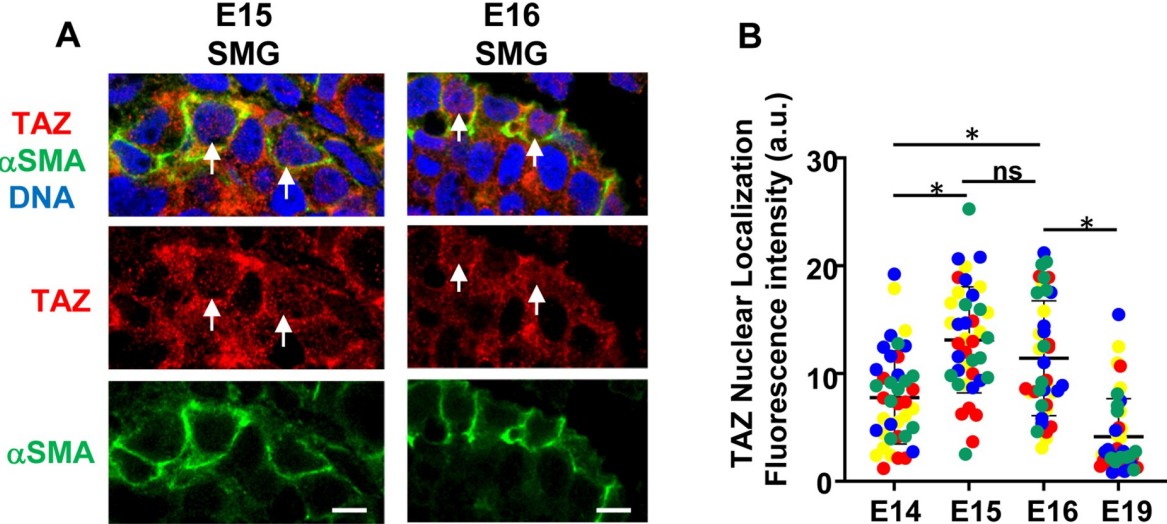

**Fig 6. Expression of α-SMA expression and TAZ in E15 and E16 SMGs. (A)** Representative confocal images of E15 and E16 embryonic SMGs immunostained for TAZ, α-SMA, and DNA. Images are a single Z-slice acquired at 63X representative of four SMGs. White arrows represent cells with α-SMA expression and with TAZ co-localization with the nuclear marker DRAQ5. White arrowhead indicates a representative cell lacking α-SMA expression and also the lack of TAZ co-localization with the nuclear marker DRAQ5. Images are single z-slices acquired at 63x. Size bar, 5 μm. **(B)** The expression of TAZ colocalized with nuclei at E14, E15, E16 and E19 was measured using ImageJ. Only cells on the basal surface of developing buds were analyzed. Plotted is the integrated fluorescence intensity of nuclear TAZ ± s.d for individual nuclei measured from four single slice confocal images acquired at 63X from two glands for each timepoint. a. u = arbitrary units. Measurements taken from each individual image are shown different colors. *p<0.05 using one-way ANOVA followed by Tukey Post-hoc test. ns = not significant.

developing salivary gland is also regulated by the LATS kinases and this regulation is needed for normal branching morphogenesis [42]. During salivary gland morphogenesis, TAZ is sequestered at cell-cell junctions by a mechanism dependent upon LATS kinases. RNAi-mediated depletion of LATS resulted in defects in branching morphogenesis, and duct formation. These phenotypes were associated with the loss of the localization of TAZ at cell-cell junctions and inappropriate TAZ activation [42]. In our studies, we examined the subcellular localization TAZ in embryonic buds at E14-E19. Consistently, we also found that TAZ was localized at cell-cell junctions in the majority of cells in epithelial buds at E15 and E16. However, in a subset of cells at the basal surface that expressed α-SMA, TAZ was co-localized in the nucleus. This localization is consistent with a role for TAZ in regulation myoepithelial differentiation in the developing salivary gland. Others have demonstrated that the interaction of cells with the basement membrane regulates the activation of YAP or TAZ depending on contexts [44–48]. Thus, the interaction of outer cuboidal epithelial cells with the maturing basement membrane of developing buds of the salivary gland may activate TAZ to promote myoepithelial differentiation. It will be important to directly demonstrate a functional role for TAZ in regulating myoepithelial differentiation during the development of the salivary gland *in vivo* and to determine whether integrin-mediated adhesion plays a role in this regulation.

Interestingly, in the mammary gland, loss of TAZ leads to defects in morphogenesis, as well as a loss of the proper balance of luminal to basal epithelial cells [41]. Additionally, TAZ was identified in a screen for transcriptional regulators that can trigger phenotypic switches between luminal and basal/myoepithelial differentiation using mammary epithelial cell lines [41]. Expression of recombinant TAZ in luminal cells led them to adopt a myoepithelial phenotype, while the inhibition of TAZ in basal/myoepithelial cells resulted in a luminal phenotype [41]. Our results are consistent with these findings. Our data also showed that conditions

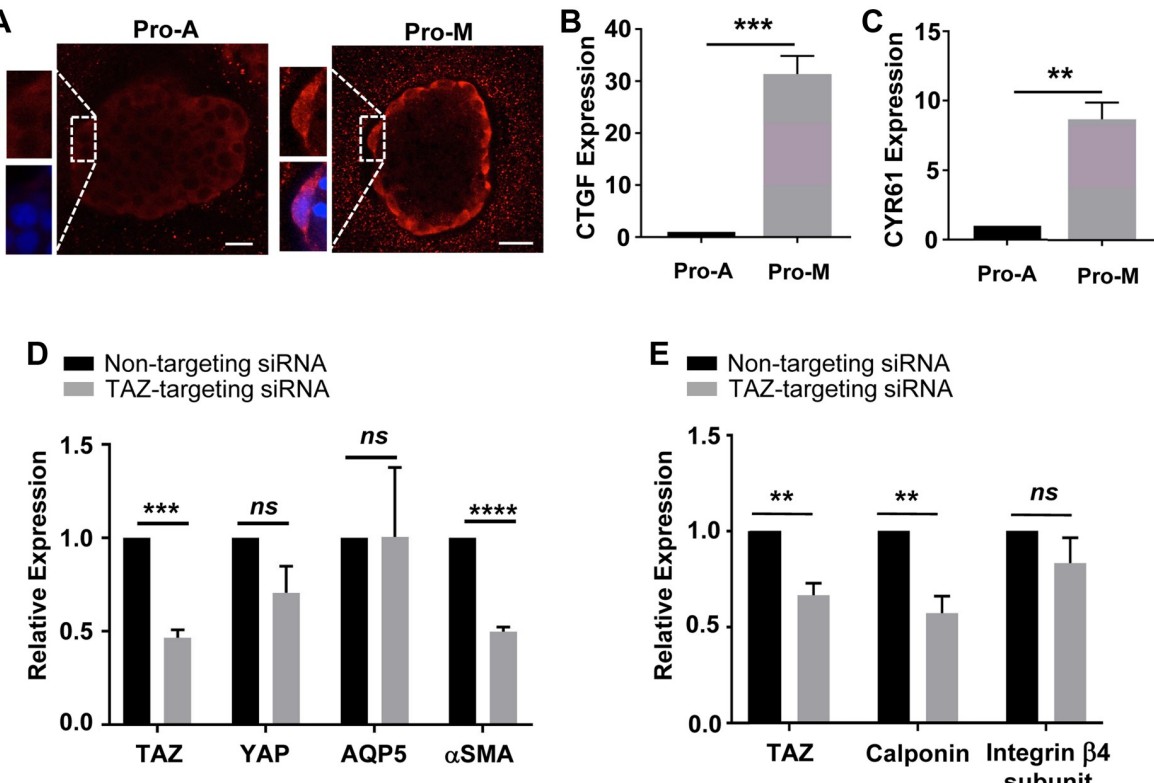

**Fig 7. mSG-PAC1 cells cultured in Pro-M medium display characteristics of myoepithelial cells.** (A) Increased expression of TAZ in the cells at the periphery mSG-PAC1 spheroids cultured in Pro-M. Representative confocal images from 2 independent experiments of mSG-PAC1 cells cultured for six days in Matrigel in either Pro-A or Pro-M medium and immunostained for TAZ (red) and DRAQ5 (pseudocolored blue). Insets represent a magnified view of nuclear staining. Images are maximum projection images of two z-slices acquired at 40X in 0.4 μm steps. Size bar, 25 μm. (B, C) Relative mRNA expression of YAP/TAZ target genes, CTGF (B) and CYR61 (C), in mSG-PAC1 cells cultured in Matrigel in either Pro-A or Pro-M medium. Expression normalized to β-actin and the expression in Pro-A medium. Data are from three independent experiments plotted as the mean ± s.e.m. analyzed by Student's T-test. \*\*p<0.01; \*\*\*p<0.001. (D) SiRNA targeting TAZ expression inhibits the expression of α-SMA. Plotted is the relative mRNA expression of YAP, TAZ, AQP5, and α-SMA in mSG-PAC1 spheroids cultured in Matrigel in Pro-M medium. Expression is normalized to β-actin and the expression in cells treated with non-targeting (NT) siRNA. Data are from 4 independent experiments and plotted as the mean ± s.e.m. analyzed by Student's T-test. *ns*, not significant, \*\*p<0.01; \*\*\*p<0.001; \*\*\*\*p<0.0001. (E) SiRNA targeting TAZ also inhibited the expression of calponin, but not the expression of the integrin β4 subunit. Plotted is the relative RNA expression of TAZ, calponin, and the integrin β4 subunit in mSG-PAC1 spheroids cultured in Matrigel and Pro-M medium. Expression is normalized to β-actin and the expression of these genes after treatment with NT siRNA. Data are from three (calponin) and four (TAZ, Integrin β4) independent experiments and plotted as the mean ± s.e.m. analyzed by Student's T-test. *ns*, not significant, \*\*p<0.01.

that promote myoepithelial differentiation inhibit the expression of the acinar marker, AQP5, but our data did not implicate TAZ in this regulation. However, a recent study demonstrated that inappropriate activation of TAZ (and not YAP) inhibited acinar differentiation [49]. The mechanism for TAZ-specific transcription is not fully understood. In the case discussed above, the ability of TAZ to regulate mammary epithelial cell fate was dependent upon its interaction with a component of the SWI/SNF chromatin remodeling complex [41]. An interestingly recent study suggests that specificity is governed by the ability of TAZ (and not YAP) to "phase separate" together with its DNA-binding partner TEAD4 and co-activators [50]. The notion of liquid-liquid phase separation of transcriptional regulators is a relatively new area of investigation into mechanisms of transcriptional control [51]. It will be interesting to determine how widely it is applicable to TAZ-specific transcription in other contexts.

Although we have identified conditions that promote the ability of mSG-PAC1 cells to recapitulate aspects of either acinar [18] or myoepithelial differentiation as reported here, we have not been able to identify culture conditions that allow mSG-PAC1 cells to differentiate into mature acinar or myoepithelial cells, or to simultaneously differentiate into both acinar and myoepithelial cells in the same Matrigel culture. It is important to note, however, that the addition of serum to Pro-A medium is not sufficient to promote mSG-PAC1 spheroids toward a myoepithelial phenotype, suggesting that components Pro-A medium may have an inhibitory effect on myoepithelial differentiation. Indeed, FGF2 has been shown to inhibit YAP/TAZ-dependent transcription in some contexts [52]. Thus, FGF2 may function to maintain a pro-acinar phenotype by suppressing TAZ activity.

Additional environmental signals are likely required to support the further differentiation of mSG-PAC1 cells. During development, both morphogenesis and differentiation of the salivary gland require signals from the surrounding mesenchyme, as well as developing nerve and endothelial networks [53, 54]. Signals from the mesenchyme have recently been shown to promote acinar differentiation [19]. It is not yet known whether myoepithelial differentiation also requires signals from surrounding cells types. The identification of environmental signals that regulate both acinar and myoepithelial differentiation may provide the needed information to allow the modelling of these differentiation events in culture.

In summary, we demonstrate that our previously characterized salivary gland epithelial cell line, mSG-PAC1, is capable of further differentiating into a myoepithelial-like layer in 3D Matrigel culture. Moreover, we identify TAZ as a regulator of myoepithelial differentiation in culture by regulating the transcription of myoepithelial genes. The expression pattern of TAZ during the differentiation of myoepithelial cells of the developing salivary gland is consistent with a similar role for TAZ *in vivo*.

## Materials and methods

### Cell culture

The establishment and characterization of the murine pro-acinar cell line, mSG-PAC1 was accomplished via the collaboration of the LaFlamme and Larsen labs as previously described [18]. These cells were maintained in a modification of the culture medium previously described for the isolation of the mammary epithelial cell line MCF10A [55, 56], which consisted of DMEM/F12 supplemented with 5% donor horse serum (Atlanta Biologicals, #S12150), 100 U/ml penicillin/streptomycin (Hyclone, #SV30010), 20 ng/ml human recombinant EGF (Gibco, #PHG0311L), 100 ng/ml Cholera Toxin (Sigma, #C8052), 2.5 μg/ml hydrocortisone (Sigma, #H0396), and 20 μg/ml human insulin (Sigma, #I9278). To induce a more pro-acinar phenotype, human recombinant EGF was replaced with 100 ng/ml bFGF/FGF2 (Peprotech, #450–33) [19]. This medium is referred to as Pro-A. To induce more myoepithelial characteristics, cells were cultured Pro-M medium, which consisted of DMEM/F12 medium supplemented with 10% FBS (Atlanta Biologicals, #S11150) and 100 U/ml penicillin/streptomycin (Hyclone, #SV30010). For three-dimensional (3-D) spheroid cultures, matrices were prepared in 8-well chamber slides (Corning, #08-774-208). Matrices consisted of 100% Matrigel (Corning, #354230, protein concentration ~10 mg/ml, endotoxin <1.5 mg/ml). Approximately, 1000 cells were plated per well and cultured for 5–7 days in medium supplemented with 2% Matrigel in either Pro-A ore Pro-M medium as indicated in the Figure Legends.

### Submandibular salivary gland explants

Murine submandibular salivary glands were dissected as previously described [57] from embryos harvested from timed-pregnant female mice (CD-1 strain, Charles River

Laboratories) at E14, E15, E16, and E19 with E0 as designated by the discovery of a vaginal plug. Embryonic salivary glands were placed on Whatman® Nuclepore filters in 35 mm Mat-Tek dishes (MatTek, #P35G-1.5-14-C) and fixed in 4% paraformaldehyde at 4°C overnight and then processed for immunostaining.

## Immunostaining

Spheroids cultured in Matrigel were fixed for 20 min in 4% paraformaldehyde, washed in 0.5% PBST, permeabilized in 0.4% Triton-X-100/1X PBS for 20 min, and then washed in 0.5% PBST before blocking 1–2 h in 20% donkey serum/PBST. Primary and secondary antibodies were incubated overnight in 3% BSA/PBST. A list of antibodies and dilutions used for immunostaining is provided in S1 Table in S1 Appendix.

Submandibular glands were washed in 0.5% PBST after overnight fixation in 4% paraformaldehyde, permeabilized in 0.4% Triton-X-100/PBS for 30 min and then washed in 0.5% PBST before blocking 1–2 h in 20% donkey serum/PBST. Primary and secondary antibodies were incubated overnight in 3% BSA/PBST. DRAQ5 (Cell Signaling) was used at a dilution of 1:1000 to detect nuclei. Coverslips and slides were mounted using SlowFade®Gold antifade mounting medium (Life Technologies, #P36930).

## RNA isolation & quantitative PCR (qPCR)

RNA was extracted with TRIzol (Ambion, ##15596026) and genomic DNA was removed with TURBO DNaseI (ThermoFisher Scientific, #AM1907) according to the manufacturers' protocols. cDNA was synthesized from 500 ng-1μg of RNA using the iScript Reverse Transcription Supermix kit (Biorad, #1708840). Equal concentrations of cDNA were used in the qPCR reactions with iQ or iTaq SYBR Green Supermix (Biorad, #170–8880; #1725120). Reactions were run in triplicate using the BioRad CFX96 Real-time C1000 Touch Thermal Cycler. Ct values were normalized to β-actin. A list of primer sequences can be found in S2 Table in S1 Appendix.

**RNA interference.** mSG-PAC1 cells were cultured in Matrigel in Pro-M medium for two days. After two days, the resulting spheroids were transfected with siRNA in OptiMEM over the course of two consecutive days using Lipofectamine RNAiMAX, according to the manufacturer's protocol. Non-targeting siRNA or siRNA targeting YAP or TAZ was transfected at a final concentration of 400 nM. Transfection medium was replaced with Pro-M medium and 2% Matrigel for 24 hours prior to processing for RNA. siRNA sequences can be found in S3 Table in S1 Appendix.

## RNA microarray and bioinformatic analysis

Microarrays were performed using RNA isolated as described above from mSG-PAC1 spheroids cultured in Matrigel in either Pro-A or Pro-M medium from three independent experiments. Samples were analyzed using the mouse Clariom® S array (Invitrogen, #902930) at the Center for Functional Genomics at The University at Albany Health Sciences Campus, Rensselaer, NY, and analyzed using the Transcriptome Analysis Console (TAC) 4.0 software from ThermoFisher. Principal component analysis (PCA) and volcano plots indicate that these culture conditions promote significant differences in gene expression (S3a, S3b Fig in S1 Appendix). Gene overlaps were computed using hallmark and curated gene sets from Molecular Signatures Database available from the Broad institute. Gene-set enrichment analysis was performed using GSEA software [58] from the Broad Institute with published scRNA-seq gene sets [37, 38]. Gene sets for myoepithelial, proacinar and acinar cells isolated from the murine submandibular salivary gland were downloaded from Supplementary Fig 6 [37]. ScRNA-seq

data sets from murine mammary glands for myoepithelial cells and luminal cells from pregnant females and luminal progenitor cells from virgin females were downloaded from Supplemental Table 5 [37, 38]. In instances where gene sets were larger than recommended for use in GSEA, subsets of genes with the highest adjusted p values were generated for the analysis. Data sets will be deposited in the NCBi Gene Expression Omnibus (GEO) data base once the manuscript is accepted for publication.

## Microscopy

Images were acquired using an inverted Nikon TE2000-E microscope with phase contrast and epifluorescence, a Prior ProScanII motorized stage, and Nikon C1 confocal system with EZC1 and NIS-Elements acquisition software, or using the Zeiss LSM 880 confocal microscope with AiryScan on an AxioObserver.Z1 motorized inverted microscope with Zeiss ZEN2.3 software. Confocal images were acquired either at 40X, 63X, or 100X, and are represented as maximum projection images or single slices, as indicated in the Figure Legends. Images were processed and analyzed using the Imaris 9 software, where indicated in the Figure Legends.

## Animal experiments

All animal experiments and procedures were performed in accordance with the Albany Medical College Institutional Animal Care and Use Committee (IACUC) regulations. In accordance with protocols approved by the Albany Medical College IACUC, mouse submandibular salivary glands (SMGs) were dissected from timed-pregnant female mice (strain CD-1, Charles River Laboratories) at embryonic day 14 (E14), 15(E15), 16 (E16) or 19 (E19) with the day of plug discovery designated as E0.

## Statistical analysis

Statistical analyses were performed using the GraphPad Prism software employing either a Student's T-test or one-way Anova followed by Tukey Post-hoc analysis as indicated in the Figure Legends. P values of $<0.05$ were deemed statistically significant.

## Supporting information

**S1 Appendix.**
(PDF)

## Acknowledgments

The authors thank the Imaging Core of Albany Medical College for assistance in the preparation of immunofluorescence images, Dr. Peter Vincent, Albany Medical College, for assistance with statistical analysis, and Deborah Moran for assistance in the preparation of the manuscript.

## Author Contributions

**Conceptualization:** Renee F. Thiemann, Scott Varney, John Lamar, Melinda Larsen, Susan E. LaFlamme.

**Data curation:** Renee F. Thiemann, Scott Varney, Nicholas Moskwa.

**Formal analysis:** Renee F. Thiemann, Scott Varney, Nicholas Moskwa.

**Funding acquisition:** Melinda Larsen, Susan E. LaFlamme.

**Investigation:** Renee F. Thiemann, Scott Varney, Nicholas Moskwa.

**Methodology:** Renee F. Thiemann, Scott Varney, Nicholas Moskwa, John Lamar.

**Project administration:** Susan E. LaFlamme.

**Resources:** Melinda Larsen, Susan E. LaFlamme.

**Supervision:** Melinda Larsen, Susan E. LaFlamme.

**Validation:** Renee F. Thiemann, Scott Varney, Nicholas Moskwa.

**Visualization:** Renee F. Thiemann, Scott Varney, Nicholas Moskwa.

**Writing – original draft:** Renee F. Thiemann.

**Writing – review & editing:** Renee F. Thiemann, Scott Varney, Nicholas Moskwa, John Lamar, Melinda Larsen, Susan E. LaFlamme.

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
