## [Decision Letter · Decision Letter 0]

10 Nov 2021

PONE-D-21-32307Regulation of myoepithelial differentiationPLOS ONE

Dear Dr. LaFlamme,

Thank you for submitting your manuscript to PLOS ONE. After careful consideration, we feel that it has merit but does not fully meet PLOS ONE’s publication criteria as it currently stands. Therefore, we invite you to submit a revised version of the manuscript that addresses the points raised during the review process.

Two experts in the field have taken part of your work and have some detailed comments that they want you to address. Please answer each reviewer point by point. Please let me know if you need extra time. I look forward to receiving the revised manuscript.

We look forward to receiving your revised manuscript.

Kind regards,

Donald Gullberg, PhD

Academic Editor

PLOS ONE

Journal Requirements:

Reviewers' comments:

Reviewer's Responses to Questions

**Comments to the Author**

1. Is the manuscript technically sound, and do the data support the conclusions?

Reviewer #1: Partly

Reviewer #2: Yes

2. Has the statistical analysis been performed appropriately and rigorously? 

Reviewer #1: Yes

Reviewer #2: I Don't Know

3. Have the authors made all data underlying the findings in their manuscript fully available?

Reviewer #1: No

Reviewer #2: Yes

4. Is the manuscript presented in an intelligible fashion and written in standard English?

Reviewer #1: Yes

Reviewer #2: Yes

5. Review Comments to the Author

Reviewer #1: Overall the manuscript offers some novel observations about the control of differentiation of cells to a myoepithelial phenotype. The core observation is that the transcription factor Taz drives changes in several genes associated with a myoeothelial phenotype. Thus with the Pro-Mmedia Figure 6 shows Taz upregulated on outer cells of the spheroids and Figure 1 shows upregulation of aSMA and Calponion.

However as one reads the manuscript, there are various points that could have been presented with greater clarity, and other areas where additional immunohistochemistry would confirm location of the changes implied by the averaged RTPCR arising from analysis of the spheroids.

1. Readers encounter the mSG-PAC1 cell line name on page 4. This is th ekey line upon which the whole paper is based. There is no explanation offered as to what the cells are which would have helped. I ran off to the Methods to find that the only explanation was a reference. Help your readers with a brief explanation and reference in the results.

2. In the paper the changes in levels of aSMA, b4, aquaporin 5, calponoin and YAP are through RNA analysis of the spheroids. Thu sthe data re the average of the whole spheroid despite the data presented clearly implying that the myoepithelial phenotype induced by the FCS containing media predominantly occurs at the outer surface of the oral submandibular spheroids. Immunohistochemistry of these proteins on the spheroids should be supplied to show where the expression of these proteins is occurring. This is especially true for integrin beta4 which is an epithelial marker not a myoepithelial marker. All the cells in the spheroid are epithelial in origin so I would expect that all the cells express this integrin. Again, the aquaporin should be higher in the inner epithelial cells of the spheroids compared with the outer layer, so show it by IHC. As presented it is not valid to imply the changes in these genes is associated with the myoepithelial phenotype if the cells with that phenotype have not been directly analysed.

3.The data on TAZ transition to the nucleus, required for transcriptional activity, should be presented more clearly by showing images of just the TAZ staining. This would show exactly where the TAZ is within cells. The purple colour (red and blue pixels co-localised) in the overlays is not wholly clear.

4. The paper is based on an RNAseq analysis of the spheroids treated with different media. 1000s of genes changed. The manuscript has chosen to pick out a handful of genes that addresses the myoepithelial phenotype, linking them with TAZ. Where is the link to the uploaded transcriptional data for the other changed genes? Isnt it possible that there are other genes that have changed that may also have implications for the myoepithelial phenotype?

Overall there are some correlations that implicate molecular processes in myoepithelial differentiation. With additional data outlined the message would be more secure.

Reviewer #2: The authors of manuscript PONE-D-21-32307 described a 3-D culture system with mSG-PAC1 salivary gland derived cell line that was useful for identifying factors involved in differentiation into the myoepithelial lineage. Supporting data were provided by IF analysis developing murine salivary gland. The authors convincingly demonstrated that TAZ (WWTR1) is and important transcription factor for myoepithelial differentiation.

Critiques:

1) Figure 1A (particularly the PRO-A condition) would benefit from addition of nuclear staining. It kind of looks like a hazy green blob, even though I am pretty sure it is an acinus.

2) Figure 1F would be substantially improved by showing every data point that was used to derive the mean w/h ratio. In its current form it is impossible to understand how the error bars are so tight, and why the statistic looks so strong. Perhaps it really is that good, but cells in nature tend towards variation in morphometrics.

3) There is no reference in the text to Figure 5C.

4) Statements about the subcellular localization of TAZ in developing glands in vivo (Figure 5) or in the mSG-PAC1 model (Figure 6) were not clearly supported by the microscopy. It was difficult to see supposed examples of localization at the cell-cell junction vs cytoplasm vs nucleus. e.g. Cytoplasmic vs Nuclear quantification would help, as would higher resolution images.

6. PLOS authors have the option to publish the peer review history of their article (what does this mean?). If published, this will include your full peer review and any attached files.

Reviewer #1: No

Reviewer #2: No

---

## [Author Response · Author response to Decision Letter 0]

19 Apr 2022

The authors thank the reviewers for their helpful comments and suggestions. We addressed these below. Please note that because of the addition of new data, old Fig.5 became too large to upload. Data in old Fig. 5 together with new data are now presented in Figs. 5 and 6. Old Fig. 6 is now Fig. 7. Additionally, we edited the text to reflect changes in response to reviewers’ comments and also to correct typographical errors and to improve readability.

REVIEWER 1

Comment 1: Readers encounter the mSG-PAC1 cell line name on page 4…. Help your readers with a brief explanation and reference in the results.

Response: In our revised manuscript, we describe how we established mSG-PAC1 cells at the end of the Introduction together with the citation.

Comment 2: In the paper the changes in levels of aSMA, b4, aquaporin 5, calponin and YAP are through RNA analysis of the spheroids… Immunohistochemistry of these proteins on the spheroids should be supplied to show where the expression of these proteins is occurring. This is especially true for integrin beta4 which is an epithelial marker not a myoepithelial marker.

Response: This data has been provided in revised Figure 1, with the exceptions of the b4 subunit, which we did not detect at the protein level. The reviewer is correct that b4 is a marker for basal epithelial cells however it is also a marker for mature myoepithelial cells (see ref #36)

Comment 3:The data on TAZ transition to the nucleus, required for transcriptional activity, should be presented more clearly by showing images of just the TAZ staining. 

Response: These images are provided in our new Figure 6. Old Figure 6 is now Fig. 7.

Comment 4: Where is the link to the uploaded transcriptional data for the other changed genes? Isn’t it possible that there are other genes that have changed that may also have implications for the myoepithelial phenotype?

Response: The link to our transcriptional data is provided in our revised manuscript. Yes, it very likely that other genes have changed that also have implications for the myoepithelial phenotype.

REVIEWER 2

Comment 1: Figure 1A (particularly the PRO-A condition) would benefit from addition of nuclear staining. It kind of looks like a hazy green blob, even though I am pretty sure it is an acinus.

Response: We have revised Figure 1 to address this and other issues.

Comment 2: Figure 1F would be substantially improved by showing every data point that was used to derive the mean w/h ratio. In its current form it is impossible to understand how the error bars are so tight, and why the statistic looks so strong. Perhaps it really is that good, but cells in nature tend towards variation in morphometrics.

Response: We have revised this graph.

Comment 3: There is no reference in the text to Figure 5C.

Response: Fig.5 has been separated into new Fig. 5 and new Fig. 6 Old Fig. 6 is now Fig. 7. All panels are now appropriately referenced in the text. 

Comment 4: Statements about the subcellular localization of TAZ in developing glands in vivo (Figure 5) or in the mSG-PAC1 model (Figure 6) were not clearly supported by the microscopy. It was difficult to see supposed examples of localization at the cell-cell junction vs cytoplasm vs nucleus. e.g. Cytoplasmic vs Nuclear quantification would help, as would higher resolution images.

Response: We agree with the reviewer that we cannot justify the localization of TAZ at cell-cell junctions. We have removed this statement from our revised manuscript. We analyzed the nuclear localization of TAZ at E14, E15, E16, and E17. This analysis is provided in our new Fig. 6. Old Fig. 6 is now Fig 7 in our revised manuscript.

---

## [Decision Letter · Decision Letter 1]

4 May 2022

Regulation of myoepithelial differentiation

PONE-D-21-32307R1

Dear Dr. LaFlamme,

Congratulations!!! We’re pleased to inform you that your manuscript has been judged scientifically suitable for publication and will be formally accepted for publication once it meets all outstanding technical requirements.

Kind regards,

Rajeev Samant

Academic Editor

PLOS ONE

Additional Editor Comments (optional):

The revised manuscript has addressed Reviewers comments. The edited/revised manuscript is much improved.

Reviewers' comments:

Reviewer's Responses to Questions

**Comments to the Author**

1. If the authors have adequately addressed your comments raised in a previous round of review and you feel that this manuscript is now acceptable for publication, you may indicate that here to bypass the “Comments to the Author” section, enter your conflict of interest statement in the “Confidential to Editor” section, and submit your "Accept" recommendation.

Reviewer #2: All comments have been addressed

2. Is the manuscript technically sound, and do the data support the conclusions?

Reviewer #2: Yes

3. Has the statistical analysis been performed appropriately and rigorously? 

Reviewer #2: I Don't Know

4. Have the authors made all data underlying the findings in their manuscript fully available?

Reviewer #2: Yes

5. Is the manuscript presented in an intelligible fashion and written in standard English?

Reviewer #2: Yes

6. Review Comments to the Author

Reviewer #2: (No Response)

7. PLOS authors have the option to publish the peer review history of their article (what does this mean?). If published, this will include your full peer review and any attached files.

Reviewer #2: No

---

## [Editor Report · Acceptance letter]

17 May 2022

PONE-D-21-32307R1 

Regulation of myoepithelial differentiation 

Dear Dr. LaFlamme:

I'm pleased to inform you that your manuscript has been deemed suitable for publication in PLOS ONE. Congratulations! Your manuscript is now with our production department. 

Kind regards, 

on behalf of

Dr. Rajeev Samant 

Academic Editor

PLOS ONE